# Genetic Structure and Diversity Analysis of Tunisian *Orobanche* spp. and *Phelipanche* spp. Using Molecular Markers

Khalil Khamassi [1,*], Zouhaier Abbes [1], Eleni Tani [2,*], Anastasios Katsileros [2], Karim Guenni [3], Mustapha Rouissi [4], Sahari Khoufi [4], Ramzi Chaabane [4,5], Demosthenis Chachalis [6] and Mohamed Kharrat [1]

[1] Field Crop Laboratory (LR16INRAT02), Institut National de la Recherche Agronomique de Tunisie (INRAT), University of Carthage, Rue Hédi Karray, 1004 Menzah 1, Tunis 2036, Tunisia; zouhaier.abbes@isste.ucar.tn (Z.A.); kharrat.mohamed@inrat.ucar.tn (M.K.)

[2] Laboratory of Plant Breeding and Biometry, Department of Crop Science, Agricultural University of Athens, 11855 Athens, Greece; katsileros@aua.gr

[3] Faculté des Sciences de Tunis, Laboratoire de Génétique Moléculaire, Immunologie et Biotechnologie (LR99ES12), Campus Universitaire, 2092 El Manar, University of Tunis El Manar, Tunis 2092, Tunisia; kguenni@yahoo.fr

[4] Agricultural Applied Biotechnology Laboratory (LR16INRAT06), Institut National de la Recherche Agronomique de Tunisie (INRAT), University of Carthage, Rue Hédi Karray, 1004 Menzah 1, Tunis 2036, Tunisia; mustapha_rssi@yahoo.fr (M.R.); sk111183@yahoo.fr (S.K.); ramzic2003@gmail.com (R.C.)

[5] National Gene Bank of Tunisia, La Charguia 1-Tunis, Boulevard Yesser Arafat, Tunis 1080, Tunisia

[6] Laboratory of Weed Science, Benaki Phytopathological Institute, Stefanou Delta 8, 14561 Kifisia, Greece; d.chachalis@bpi.gr

* Correspondence: khalilkhamassi9@gmail.com (K.K.); etani@aua.gr (E.T.)

**Abstract:** Broomrapes (*Orobanche* and *Phelipanche* spp.) are non-achlorophyllous parasitic plants belonging to the Orobanchaceae family, with some species evolving to infest agricultural crops, causing substantial economic losses. This study focuses on *Orobanche* and *Phelipenche* species prevalent in Tunisia, particularly *Orobanche crenata*, *Orobanche foetida* and *Phelipanche ramosa*, which pose a significant threat to legume crops and other agronomically important plants. These parasitic species cause severe damage before their aboveground appearance, making early detection and management crucial. Successful breeding programs targeting their hosts necessitate a comprehensive understanding of the genetic variability within different broomrape populations. A plethora of molecular markers, including RAPD, ISSR, AFLP, SSR and SNPs, were employed to evaluate the genetic diversity of *Orobanche* spp., mainly in Mediterranean countries. This research seeks to analyze the genetic variability and structure of thirty-four (34) Tunisian *Orobanche* and *Phelipanche* populations infesting various crops and wild plants. The results demonstrated a higher genetic differentiation within populations rather than between populations and no clear differentiation based on the geographic origins of the populations. By measuring the genetic diversity of a large number of broomrape populations that affect both wild species and crops, this study aims to support efforts toward establishing effective management approaches.

**Keywords:** *Orobanche foetida*; *Orobanche crenata*; *Phelipanche ramosa*; population structure; genetic diversity

## 1. Introduction

The non-chlorophyllous parasitic plants known as Broomrapes (*Orobanche* and *Phelipanche* spp.) belong to the Orobanchaceae family [1,2]. There are approximately 150 recorded species of broomrapes, most of which infest wild plants in natural habitats without causing economic problems; few of them have become serious weeds that infest important crops as obligate holophrastic root weeds. In this context, the most damaging broomrapes are *O. crenata*, *O. cernua*, *O. foetida*, *O. cumana*, *O. minor*, *P. aegyptiaca* and *P. ramosa*, which cause serious problems, and even the total loss of production, in important dicot crops in African,

Asian and European countries; these species are constantly expanding into new areas, demonstrating their ability to evolve, thus expanding their host range [3].

Indeed, having a better understanding of the genetic evolution, differentiation and spread of these parasites is very urgent, as broomrapes are becoming a real threat to food security. In addition, the controversial phenotypic classification of broomrapes, which is a very hard task due to the reduced number of phenotypic descriptors, is leading taxonomists to errors. For these reasons, the use of molecular tools is necessary to identify and differentiate properly different broomrape species. In Tunisia, the dominant broomrape species are *O. crenata*, *O. foetida* and *P. ramose*, with no accurate estimation of their impact on Tunisian agriculture even though 5000–70,000 (ha) hectares of legume crops could be infected [4]. In order to overcome this problem, some farmers have been replacing sensitive legume crops with others, such as sunflowers, oilseed rape and garlic (personal observation). However, the above strategy is not sound since the first infestation of *O. cumana* in sunflowers has been reported, particularly in the most infected regions (i.e., the Beja region) [5].

In this context, a recent large screening of sunflower collection in infested fields, inoculated pots and square rhizotrons infected by *O. cumana* shown phenotypic parasitism variability, from sensitivity to partial resistance [6]. As such, in Tunisia, the recent efforts made by seed companies and some farmers to promote oil seed rape or canola will face a serious problem of infestation by *P. ramosa* [7].

Yield losses due to *Orobanche* spp. and *Phelipanche* spp. infestation range from 20–80% [8]. *O. crenata* is mostly spread in the western–northern, northern, and central–eastern regions of the country, especially in faba bean crops, where losses caused by the parasite can reach up to 97% [9], whereas *O. foetida*, which is an emerging threat for faba beans, is mainly found in northern and northern–western parts of Tunisia. Finally, *P. ramosa* is reported to attack legumes, tobacco and many vegetable crops, as well as oil seed rape [7].

Broomrapes cause severe damage even before their appearance aboveground. Therefore, most crop losses may occur before the infestation is clearly observed. In the literature, many different strategies have long been proposed, such as hygiene and prevention measures, the use of selective herbicides, biological control, soil treatment with fumigants, sun disinfection and trap crops. Nevertheless, they have not provided sufficient controls representing poor solutions in real-field, large-scale conditions. The key strategy, therefore, is to develop resistant crops via breeding, supported by a rapid, accurate and reliable diagnostic method for the detection of the tissues or spores of the pest in soil samples from infected crops [10].

To develop successful breeding programs toward crop tolerance or resistance to parasitic weeds, a strong emphasis should be placed on investigating and identifying the genetic variability within and among broomrape populations since their virulence depends on their genetic structure and high diversity [11,12]. Given the controversial phenotypic classification of broomrapes and the reduced number of phenotypic descriptors, the use of molecular tools is necessary to properly identify and differentiate different broomrape species. Modern breeding efforts are always indicating the use of molecular markers to thoroughly examine the diversity of the genetic material in use [13,14]. Several studies have been carried out in recent years to analyze the genetic diversity of *Orobanche* spp./*Phelipanche* spp. using molecular markers, mostly in countries of the Mediterranean region (i.e., Spain, Tunisia, Morocco, Algeria, Turkey) but also in other countries such as Ethiopia, Iran, Bulgaria, etc.

The most popular molecular markers are RAPD [15–18], ISSR [19–21] and AFLP [22,23] used separately or in combinations [2,24]. SSR markers are currently being developed and have provided useful information in several cases [12,25,26]. Moreover, in recent years, the utility of more advanced molecular techniques, such as high-resolution melting analysis (HRM) [27] and SNP coupled with sequencing [11,28], have been recognized as the most appropriate means of precisely characterizing and distinguishing different broomrape

species. However, these techniques cannot be applied in large-scale screening experiments because of cost limitations.

Certain problems are highlighted when working with broomrape genetic variability screening. These species do not form leaves and have complex vegetative organs that lead to many errors and confusion; as such, there are no standardized descriptors for the description or classification of these species. Indeed, the majority of studies dealing with the identification or genetic diversity of broomrapes tend to be very objective and based on the morphology/characteristics of the flowers or seeds with no sound discrimination criteria. Furthermore, the classification of *Orobanche* spp. is further complicated by the inherent variability and interaction of these species with their hosts [29].

The objective of this study is to document the genetic variability and structure of thirty-four (34) populations of *O. crenata*, *O. foetida* and *P. ramosa* and wild species from the northwestern region of Tunisia, which is the main grain legume cultivation area, among other crops.

## 2. Materials and Methods

### 2.1. Sampled Broomrapes and Their Geographic Localization

The broomrape samples consisted of the spikes (stem and flowers) of thirty-four *O. crenata*, *O. foetida* and *P. ramosa* populations affecting different crops and wild species from the northern and central prospected regions of Tunisia. The geographic localization of the sampled broomrapes and their respective hosts are summarized in Table 1 and Figure 1.

**Table 1.** Sampling localization of weed parasites and their host plant species.

| id | Code | Population | Parasite Specie | Host Plant | Geographical Origin | GPS |
|---|---|---|---|---|---|---|
| 3 | OC-3 | pop1 | *O. crenata* | Geranium | Amdoun Beja | 36°43′28.1″ N 9°07′23.6″ E |
| 7 | OC-7 | pop1 | *O. crenata* | Milk thistle (*Silybum marianum*) | Fritissa Farm Bizerte | 36°55′21.6″ N 9°36′46.3″ E |
| 8 | OC-8 | pop1 | *O. crenata* | *Lathyrus sativus* | Ariana | 36°55′43.3″ N 10°02′38.9″ E |
| 9 | OC-9 | pop1 | *O. crenata* | Faba bean *Vicia faba* | Rasjbal_Bizerte | 37°13′30.6″ N 10°09′03.0″ E |
| 10 | OC-10 | pop1 | *O. crenata* | Faba bean *Vicia faba* | Rasjbal_Bizerte | 37°12′56.3″ N 10°08′45.9″ E |
| 12 | OC-12 | pop1 | *O. crenata* | The sweet pea *Lathyrus odoratus* | Rn7 Tunis | 36°50′06.6″ N 10°03′25.7″ E |
| 14 | OC-14 | pop1 | *O. crenata* | Faba bean *Vicia faba* | Abida Kairouan | 35°35′31.5″ N 10°00′26.1″ E |
| 15 | OC-15 | pop1 | *O. crenata* | Milk thistle (*Silybum marianum*) | Megrine Ben Arous | 36°46′15.2″ N 10°14′24.7″ E |
| 17 | OC-17 | pop1 | *O. crenata* | Milk thistle (*Silybum marianum*) | Farm 1 Kairouan | 35°36′13.6″ N 9°53′43.5″ E |
| 18 | OC-18 | pop1 | *O. crenata* | Lathyrus sativus | Ariana | 36°56′04.9″ N 10°02′49.6″ E |
| 19 | OC-19 | pop1 | *O. crenata* | Milk thistle (*Silybum marianum*) | Elbaten Kairouan | 35°71′55.80 N 10°00′45.33″ E |
| 20 | OC-20 | pop1 | *O. crenata* | Faba bean *Vicia faba* | Sidi Ali Kairouan | 35°35′00.2″ N 9°54′01.5″ E |

**Table 1.** *Cont.*

| id | Code | Population | Parasite Specie | Host Plant | Geographical Origin | GPS |
|---|---|---|---|---|---|---|
| 21 | OC-21 | pop1 | *O. crenata* | couch grass *Elymus repens* | El Menzah 4 Tunis | 36°50′16.6″ N 10°11′01.3″ E |
| 22 | OC-22 | pop1 | *O. crenata* | Milk thistle (*Silybum marianum*) | Abida Kairouan | 35°35′30.8″ N 9°58′47.4″ E |
| 2 | OF-2 | pop2 | *O. foetida* | Faba bean *Vicia faba* | Amdoun Beja | 36°43′28.1″ N 9°07′23.6″ E |
| 4 | OF-4 | pop2 | *O. foetida* | Chickpea *Cicer arietinum* | Amdoun Beja | 36°43′28.1″ N 9°07′23.6″ E |
| 5 | OF-5 | pop2 | *O. foetida* | *Lathyrus sativus* | Amdoun Beja | 36°43′28.1″ N 9°07′23.6″ E |
| 6 | OF-6 | pop2 | *O. foetida* | Faba bean *Vicia faba* | Oued Beja Beja | 36°44′07.2″ N 9°13′33.4″ E |
| 13 | OF-13 | pop2 | *O. foetida* | Chickpea *Cicer arietinum* | Amdoun Beja | 36°47′59.8″ N 9°06′29.6″ E |
| 23 | OF-23 | pop2 | *O. foetida* | Faba bean *Vicia faba* | Oued Beja Beja | 36°44′07.2″ N 9°13′33.4″ E |
| 24 | OF-24 | pop2 | *O. foetida* | Chickpea *Cicer arietinum* | El Hamrounia Beja | 36°43′24.2″ N 9°07′13.4″ E |
| 25 | OF-25 | pop2 | *O. foetida* | Faba bean *Vicia faba* | Farm Hamrounia Beja | 36°43′04.2″ N 9°07′15.8″ E |
| 26 | OF-26 | pop2 | *O. foetida* | *Calicotome spinosa* | Borj Cedria Benaours | 36°42′25.3″ N 10°23′55.5″ E |
| 27 | OF-27 | pop2 | *O. foetida* | Lentil *lens culinaris* | Lafareg Beja | 36°39′42.9″ N 9°05′25.4″ E |
| 31 | OF-31 | pop2 | *O. foetida albinos* | Faba bean *Vicia faba* | Oued Beja, Beja | 36°44′07.2″ N 9°13′33.4″ E |
| 32 | OF-32 | pop2 | *O. foetida* | Chickpea *Cicer arietinum* | Hammam Siala Beja | 36°39′39.6″ N 9°09′01.7″ E |
| 33 | OF-33 | pop2 | *O. foetida* | *Medicago truncatula* | Lafereg Beja | 36°39′42.9″ N 9°05′25.4″ E |
| 34 | OF-34 | pop2 | *O. foetida* | *Medicago scutelata* | Lafereg Beja | 36°39′42.9″ N 9°05′25.4″ E |
| 1 | PR-1 | pop3 | *P. ramose* | Couch grass *Elymus repens* | Fritissa Bizerte | 36°55′21.6″ N 9°36′46.3″ E |
| 11 | PR-11 | Pop3 | *P. ramose* | Tomato | Bargou-Siliana | 36°05′27.4″ N 9°34′22.1″ E |
| 16 | PR-16 | pop3 | *P. ramose* | Oilseed rape *Brassica napus*: | Menchar Beja | 36°44′08.0″ N 9°14′19.7″ E |
| 28 | PR-28 | pop3 | *P. ramose* | Couch grass *Elymus repens* | Ras Jebel_Bizerte | 37°13′34.5″ N 10°09′02.0″ E |
| 29 | PR-29 | pop3 | *P. ramose* | Oilseed rape *Brassica napus*: | Bizerte | 37°08′49.0″ N 9°59′36.1″ E |
| 30 | PR-30 | pop3 | *P. ramose* | Oilseed rape *Brassica napus*: | Bizerte Sidimechreg | 37°01′19.0″ N 9°39′40.3″ E |

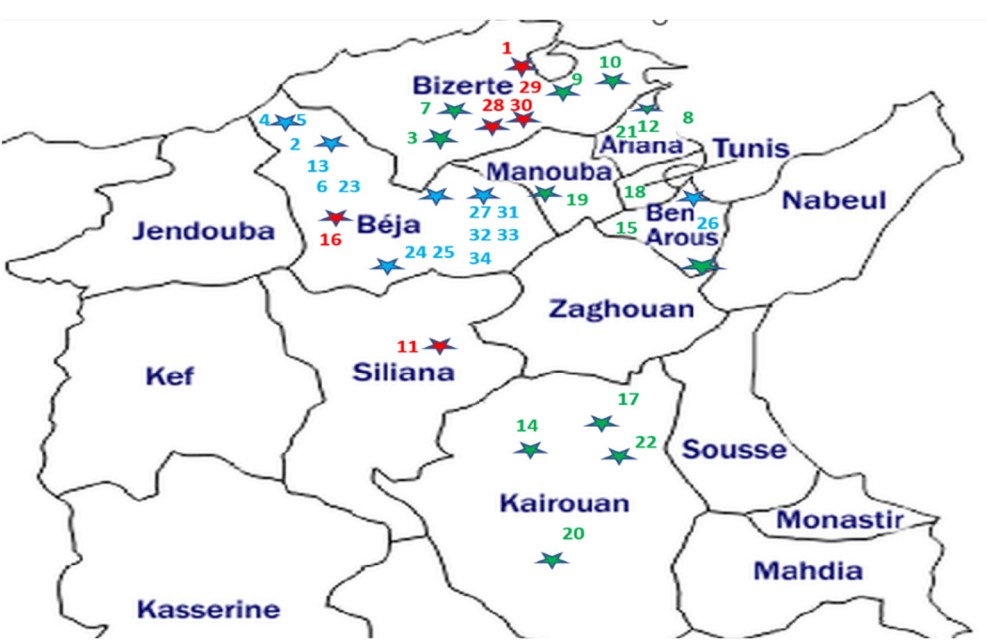

**Figure 1.** Geographical location map of the studied parasitic weeds affecting food legumes and weeds in Tunisia. The host sample IDs are from Table 1. Parasite samples—blue: *O. foetida*; green: *O. crenata*; red: *P. ramosa.* Each dot star is indicating the sampling site with its parasite corresponding color.

*2.2. DNA Extraction Protocol*

Whole genomic DNA was taken from the fresh floral buds (100 mg per sample) of each *Orobanche* sp. and *Phelipanche* sp. Each sample was a unique spike found attached to the respective host. We used a modified protocol [30] including 2% cetyltrimethylammonium bromide (CTAB) buffer (Table 2). DNA quantity/µL and quality were evaluated via both spectrophotometric absorbance (260 nm and 280 nm) and 1% ethidium bromide-stained agarose gel electrophoresis.

**Table 2.** DNA extraction modified minipreparation composition [31,32].

| DNA Micro-Prep Solution | For 100 Samples |
|---|---|
| 2.5× DNA Extraction buffer CTAB 2X | 25 (mL) |
| 2.5× lysis buffer | 25 (mL) |
| 1 × 5% Sarkosyl (Sigma Aldrich, Baden-Württemberg, Germany) | 10 (mL) |
| Sodium Bisulfite (Sigma Aldrich, Baden-Württemberg, Germany) | 0.2 g |
| RNase Dnase-free (Vivantis) | 60 mL |
| Total volume to complete up to with Nuclease-free $H_2O$ (pH = 8) | 100 (mL) |

*2.3. Primer and Polymorphic Chain Reaction (PCR) Condition*

Nine-base-long RAPD primers (OPERON Technologies, Louisville, KY, USA) were analyzed (Table 3). These primers were carefully chosen because of their high polymorphism and repeatability in studies with *O. crenata* and *O. foetida* [18,31–34], such as OPF-03 or our unpublished preliminary results.

**Table 3.** Used RAPD polymorphic primer sequences.

| Name of Primer | Primer Sequence (5′-3′) | References |
|---|---|---|
| OPF-03 | CCTGATCACC | |
| OPJ-10 | AAGCCCGAGG | |
| OPJ-01 | CCCGGCATAA | [31,34] |
| OPE-17 | CTACTGCCGT | [31,33,34] |
| OPD-20 | ACCCGGTCAC | |
| OPD-10 | GGTCTACACC | |
| OPH-13 | GACGCCACAC | |
| OPG-12 | CAGCTCACGA | |
| OPG-14 | GGATGAGACC | |

The RAPD-PCR amplifications were performed in a 25 μL volume mix for each sample consisting of 12.5 μL of a standardized 10X PCR ready-to-use master mix (Promega, MW, USA), 2 μL of 30 ng/μL template genomic DNA and GoTaq DNA polymerase (Promega, WI, USA) following Table 4. Amplification was executed via a standard RAPD-PCR program using an ''Simpliamp'' (Applied Biosystems, CA, USA) 96-Well Thermal Cycler (Table 5).

**Table 4.** PCR reaction mix composition.

| Component | | Initial Concentration | Final Concentration | Volume/Sample (μL) |
|---|---|---|---|---|
| 2X PCR master mix | $MgC_{l2}$ | 3 mM | | 12.5 μL |
| (Promega, USA) | dNTPs | 400 μM each dNTP | 1× | |
| Primer Operon | | 2 μM | 0.1 μM | 0.5 |
| GoTaq DNA polymerase (Promega, USA) | | 5 unit/μL | 1 unit/μL | 0.2 |
| DNA | | - | 30 ng/uL | 2 |
| $H_2O$ nuclease-free | | | | 11.8 |

**Table 5.** PCR program for amplification of RAPD primers.

| Number of Cycles | | Temp (°C) | Time |
|---|---|---|---|
| 1 | Initial denaturation | 94 | 4 min |
| 35 | Denaturation | 95 | 25 s |
| | Hybridization | Annealing t°(c) | 25 s |
| | Extension | 72 | 1 min |
| 1 | Post-extension | 35 | 5 min |

Amplicons were visualized via electrophoresis with 3% agarose gels with ethidium bromide-stained DNA. The sizing of amplicons was performed via comparison to a standard DNA ladder, 100 bp (Promega, USA). The RAPD dominant-marker-amplified bands were scored as 0 (absent) or 1 (present) in the scoring matrix.

*2.4. Data Analysis*

The Rp index (resolving power index) was calculated to estimate the ability of the nine RAPD primers to differentiate between genotypes following the formulation below [32]:

$$Rp = \sum I_b$$

$$I_b = 1 - (2 \times |0.5 - p|)$$

$I_b$: amplicon's informativeness; p: percentage of individuals containing amplicon I.

Moreover, the PIC index (polymorphism information content) was determined to assess the efficacy of each RAPD primer in identifying polymorphic loci both within and across populations using the following equation:

$$PIC = 1 - \sum Pi^2$$

Pi is the ith allele's frequency [35].

The usefulness per marker was evaluated indirectly via the indices: PPB (the proportion of polymorphic bands or amplicon), MI (marker index) and EMR (effective multiplex ratio), as described by [36]. The binary data matrix resulting from the RAPD polymorphism was processed through the PopGene software (Version 1.31) based on the assumption of Hardy–Weinberg equilibrium [37]. This analysis provided a structure of measurement for the population's genetic diversity degree, including Nei's genetic diversity index (H) [38], PPB and the Shannon (I) information index. Nei's gene diversity statistics [39] were utilized to determine the amount of gene or interpopulation differentiation for several loci ($G_{ST}$). The method outlined by [40] was employed to estimate gene flow (Nm) following the formula below:

$$N_m = \frac{0.5 \times (1 - G_{ST})}{G_{ST}}$$

Furthermore, a molecular variation analysis within broomrape species (AMOVA) was conducted to determine RAPD's statistical variance components. This analysis divided the variation both within and between species using the GenAlEx 6.5 software [41]. A non-parametric test was used to estimate *p*-value significance after 1000 random permutations. The neighbor-joining (NJ) method was selected to construct a dendrogram. To assess the dependability of the clusters, a bootstrapping analysis was carried out with 1000 resamples using the DARwin software, version 5.0.158 [42].

The diversity and differentiation between individuals (genetic relationships) were evaluated through principal coordinate analysis (PCoA) using PAST version 2.17c [43].

The genetic structure of the population was inferred through Bayesian methods of clustering implemented in STRUCTURE version 2.3.4 [44].

An ad hoc method to assess the probable number of clusters, K, based on ΔK was developed by [45]. ΔK is a statistic used to determine the optimal number of genetic clusters (K) in a population when performing Bayesian clustering analysis. The formula for ΔK is as follows:

$$\Delta K = |L'(K) - L(K - 1)| / s(K)$$

where $L'(K)$ represents the mean likelihood of K; $L(K - 1)|$ represents the mean likelihood of K minus one (the previous K value); and $s(K)$ represents the standard deviation of the likelihood values of K.

The K with the highest ΔK is considered the optimal number of genetic clusters. This statistic is valuable for determining the genetic structure of populations and understanding how individuals group together based on their genetic data.

Within STRUCTURE, we operated under an admixture model, considering the prior data from the sampling site. In total, 10 repetitions were executed to the respective potential values of K (ranging from K = 1 to K = 6) consisting of 70,000 repetitions and 100,000 burn-in steps. The online tool STRUCTURE Harvester was used as an easier way to distinguish the sum of genetically similar groups (K) that best fit the data that we employed [44,45]. To ensure a better arrangement of independent runs, we utilized CLUMPP version 1.1.2 [46], employing the "Greedy" algorithm. This involved 10,000 random input sequences and an extra 10,000 repetitions that provided the pairwise similarity score (H') of the runs. Eventually, the Distruct version 1.1 visualization tool [47] was used for cluster representation.

## 3. Results

### 3.1. RAPD Polymorphism

We evaluated the capability of the ten selected RAPD Operons (B, D, E, F, G, H and J) for the random amplification of selected genotypes using PCR. The characteristics of these primers, when applied to the thirty-three genotypes tested, are detailed in Table 6. Out of the 10 primers used, 98 bands were recorded, with 97 (or 98.98%) of them being polymorphic. The sum of polymorphic amplicons using different RAPD primers varied from 6 (OPG12 and OPG14) to 15 (OPJ01). All primers had 100% polymorphism except the OPD20 primer (88%) (Table 6). The primer's Rp value varied from 2.26 (OPG12) to 6.82 (OPJ01); meanwhile, all primers registered high PIC, varying from 0.79 (for OPG12) to 0.92 (for OPJ01), with an average of 0.87.

**Table 6.** Polymorphism features of the ten RAPD primers for the thirty-four *Orobanche* sp. and *Phelipanche* sp. samples.

| Primer Code | TNB | NPB | PPB (%) | Rp | PIC | EMR | MI |
|---|---|---|---|---|---|---|---|
| OPB03 | 13 | 13 | 100 | 4.32 | 0.91 | 13.00 | 11.80 |
| OPD10 | 11 | 11 | 100 | 4.88 | 0.88 | 11.00 | 9.71 |
| OPD20 | 8 | 7 | 88 | 4.26 | 0.86 | 7.00 | 6.00 |
| OPE17 | 8 | 8 | 100 | 3.85 | 0.84 | 8.00 | 6.76 |
| OPF03 | 9 | 9 | 100 | 4.44 | 0.87 | 9.00 | 7.87 |
| OPG12 | 6 | 6 | 100 | 2.26 | 0.79 | 6.00 | 4.74 |
| OPG14 | 6 | 6 | 100 | 2.56 | 0.81 | 6.00 | 4.84 |
| OPH13 | 11 | 11 | 100 | 5.26 | 0.90 | 11.00 | 9.86 |
| OPJ01 | 15 | 15 | 100 | 6.82 | 0.92 | 15.00 | 13.79 |
| OPJ10 | 11 | 11 | 100 | 5.74 | 0.88 | 11.00 | 9.69 |
| Total | 98 | 97 | | 44.41 | | 97.00 | 85.06 |
| Mean | 9.80 | 9.70 | 98.98 | 4.44 | 0.87 | 9.70 | 8.51 |

TNB = total number of amplicons; NPB = total number of polymorphic amplicons; PPB = total percentage of polymorphic amplicons; Rp = marker resolving power; PIC = markers' polymorphism information content; EMR: effective multiplex ratio; and MI: marker index.

### 3.2. Genetic Diversity and Structure Explored Using RAPD Markers

*O. crenata* has the highest diversity estimators with I = 0.483 and PPB = 91.84%, whereas *P. ramosa* has the lowest: I = 0.391 and PPB = 72.45% (Table 7). The GST provided a value of 0.207, and the AMOVA showed that 70.31% of the total genetic variability happened within species and 29.69% between species (Table 8).

The low frequency of genetic variability between the broomrape species is supported by the high gene flow (Nm = 1.912).

To calculate allelic frequencies in the absence of genotypic information, which is the case for the markers studied (RAPD), we assumed the following:

- Alleles from different loci never co-migrate in a gel;
- Each locus has bi-allelic determinism.

The two molecular phenotypes' presence, [A], and absence, [a], of a fragment actually correspond to three genotypes: (AA), (Aa) and (aa) (heterozygotes (Aa) and homozygotes (AA) represent the same phenotype, [A]). In this sense, estimates of genetic diversity based on RAPD markers were carried out with reference to the work of Lynch and Milligan (1994).

**Table 7.** Genetic variation statistics and Shannon's diversity estimation for three broomrapes species.

| Population | Polymorphic Bands | | Na | Ne | H | I |
|---|---|---|---|---|---|---|
| | No. | PPB (%) | | | | |
| *O. crenata* | 90 | 91.84 | 1.918 | 1.566 | 0.325 | 0.483 |
| *O. foetida* | 79 | 80.61 | 1.806 | 1.441 | 0.261 | 0.395 |
| *P. ramosa* | 71 | 72.45 | 1.724 | 1.454 | 0.263 | 0.391 |
| Mean | 80 | 81.63 | 1.816 | 1.487 | 0.283 | 0.423 |
| Total | 97 | 98.98 | 1.989 | 1.612 | 0.358 | 0.533 |

No: Number of polymorphic amplicons; PPB: the percentage of polymorphic amplicons (%); Na: observed number of alleles; Ne: effective number of alleles; H: Nei's genetic diversity index; I: Shannon's diversity index. The observed number of alleles (Na) is the actual number of alleles that we found in broomrape populations. The effective number of alleles (Ne) is the number of equally frequent alleles that it would take to achieve the same expected heterozygosity as in our studied populations. The effective number of alleles (Ne) is, in general, lower than the observed number of alleles (Na).

**Table 8.** AMOVA of Tunisian broomrapes.

| Source of Variation | d.f. | Sum of Squares | Mean Squares | % Variation | *p*-Value |
|---|---|---|---|---|---|
| Between Species | 2 | 175.269 | 87.634 | 29.69% | <0.001 |
| Within Species | 31 | 492.143 | 15.876 | 70.31% | <0.001 |
| Total | 33 | 667.412 | | $\Phi st = 0.297$ | |

d.f.: degrees of freedom; *p*-value: significance after 1000 random changes; Φst: differences between species.

### 3.3. Neighbor-Joining Method and Principal Coordinate Analysis

Both the NJ method and the PCoA, presented in Figures 2 and 3, respectively, are used to depict the genetic relationships between the studied broomrape species. The NJ method clearly shows three major groups corresponding to the three studied broomrape species. However, a genetic similarity is noticeable between the genotypes of *P. ramosa* and the genotypes of the species *O. crenata*. Meanwhile, the genotypes of *O. foetida* differ significantly from the other two species. The PCoA reveals the same observations with a slight overlap between *P. ramosa* and *O. crenata*, (mainly due to the similarity of their hosts) [8], in particular, the genetic closeness of the two PR-11 and PR-16 genotypes to OC-3 and OC-22.

### 3.4. Tunisian Broomrape's Genetic Structure

The genetic structure of the three studied broomrape species was analyzed based on a model comprising two to three clusters (K = 2, K = 3 and K = 4). The ad hoc measure, derived from the second-order rate of variation of the likelihood function (ΔK) [45], indicated a primary clustering at K = 2 for the studied Tunisian broomrape (ΔK = 139.06) and a further subgrouping at K = 4 (ΔK = 6.66).

Each vertical bar represents an individual sample, segmented into K colors. Each colored section characterizes the estimated level of that individual's association with a specific genetic cluster.

The Clumpp program generated a permuted average Q-matrix after ten STRUCTURE runs, which provided the highest H at K = 2 and K = 4, equal to 0.997 and 0.979, respectively. This suggests reliable results for both models (Figure 4). Based on the K = 2 model, the broomrape genotypes were categorized into two genetically distinct groups or metapopulations as determined from the STRUCTURE analysis: Cluster 1 (blue) included genotypes of two species: *O. crenata* and *P. ramosa*. Cluster 2 (green) included *O. foetida* genotypes. We note that genotypes of *foetida* were assigned with over 70% probability to cluster 2 (green), whatever the model, which justifies the genotypic specificity of *O. foetida* compared with the broomrapes studied. From K = 3, we note the appearance of a third cluster (red),

bringing together three genotypes of the *P. ramosa* species, in this case, PR-28, PR-29 and PR-30, which differ significantly with an assignment probability greater than 80%, unlike PR-1, PR-11 and PR-16, which show significant genetic similarity with *O. crenata* (cluster blue).

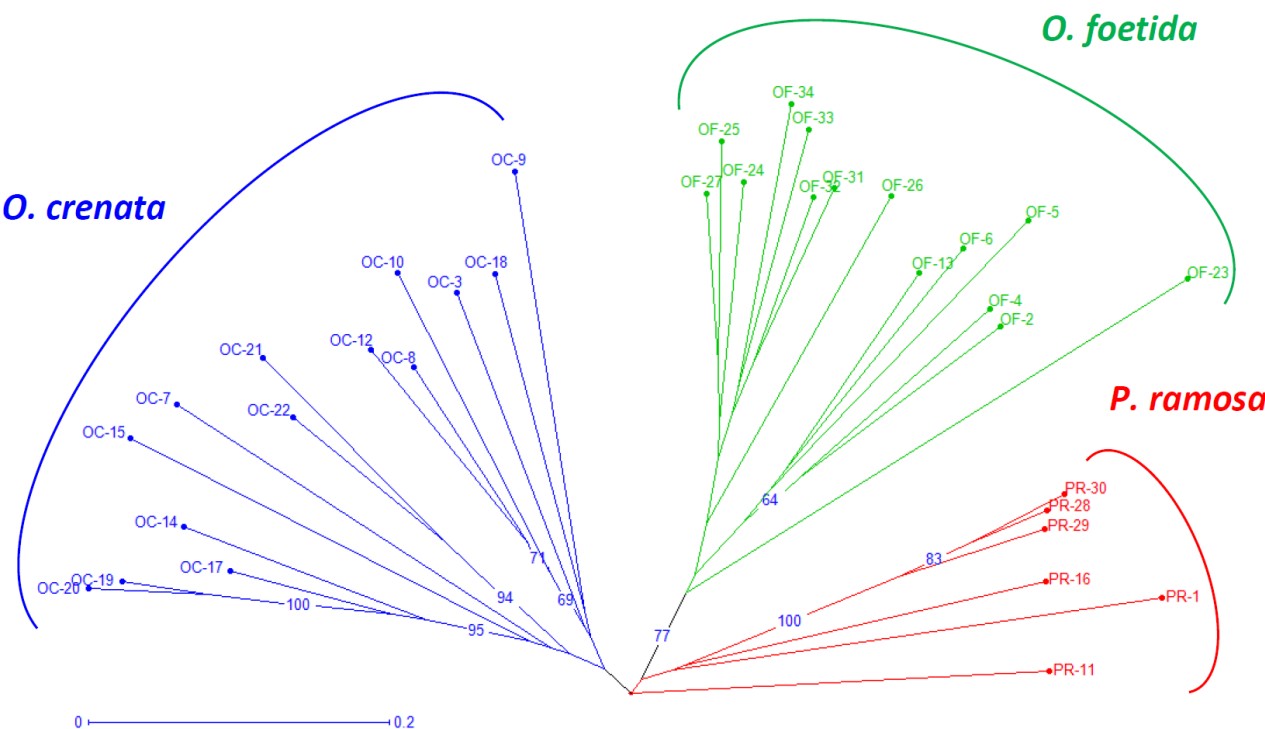

**Figure 2.** Neighbor-joining tree of broomrape derived from RAPD markers. The neighbor-joining tree displays genetic distances, rooted following Jaccard's genetic distances. Numbers at the branches are percentages that indicate the degree of 1000 bootstrap replicates. Branches collapse when they have less than 60% support. Branches of different color group different species. The coding of the different populations is presented according to Table 1.

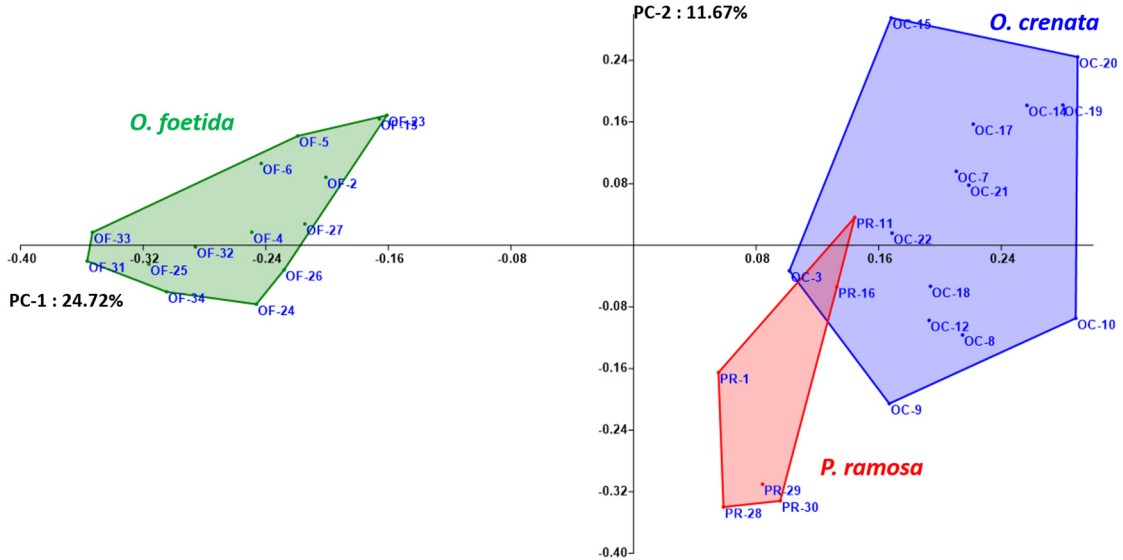

**Figure 3.** Principal coordinate analysis (PCoA) of broomrape derived from RAPD markers. Different colors indicate different species.

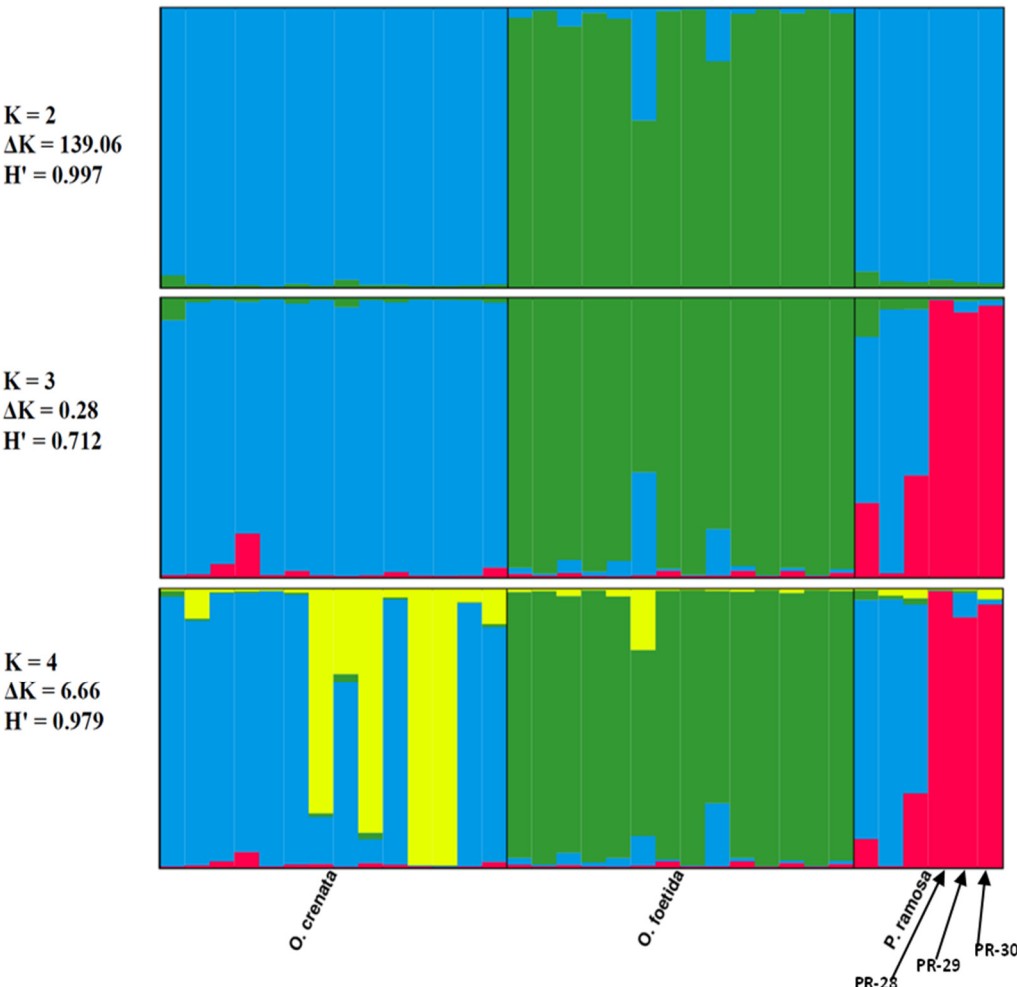

**Figure 4.** Results of the STRUCTURE analysis's genetic clustering (N = 34). Different colors indicate different clusters.

At K = 4, we note the appearance of a sub-cluster (yellow) grouping four genotypes of *O. crenata*, which seems to indicate a subdivision, but we cannot confirm this observation, because the K = 4 model is not very stable compared with K = 2, especially since the neighbor-joining method and PCoA do not show a subdivision within *O. crenata*.

## 4. Discussion

The present study is an original genetic diversity and structure analysis based on the dominant RAPD molecular markers of samples of *O. foetida*, *O. crenata* and *P. ramosa* from different hosts and regions in Tunisia. RAPD markers were chosen based on their high polymorphism and discrimination capacity as identified in our own unpublished optimizations.

Moreover, the genome of these species has not yet been sequenced, and there are no available standard molecular markers or kits that have been published to study the three species' diversity altogether. RAPD was previously used to investigate broomrape [15,18,48,49] genetic diversity in Tunisian populations [31,49], and RADseq was used to study the genetic diversity of Tunisian *O. foetida* populations [11].

The RAPD markers applied in this study clearly showed their efficiency in revealing the polymorphism between *Orobanche* and *Phelipanche* species and individuals. Indeed, the PIC ranged between 0.79 and 0.92. These results agree with those of [18], who reported the effectiveness of RAPD markers in revealing the polymorphism of varied populations of *Orobanche* spp. affecting wild hosts in Spain.

The highest PPB was monitored for *O. crenata*, followed by 80.6% for *O. foetida*, while the lowest was detected for *P. ramosa* (72.45%). These results were supported by the AMOVA, which revealed 70.31% genetic variability within species and only 29.69% between species, in agreement with previous results (75.4%) showing internal variation in Tunisian and Spanish *O. foetida* populations [34,49]. Comparable results were found in a study undertaken by [25] (the highest variability occurred between individuals as opposed to within populations based on SSR markers that screened a significant *O. crenata* population number in Ethiopia). In our study, there was a high gene flow measured between species (Nm = 1.91). This result could be based on the high level of outcrossing (71%) of *O. crenata* [2] due to its flower morphology, with large low lips that serve as a platform for pollinators. In this context, it is well documented that such plant species possess a low rate of diversity among populations compared to self-pollinated ones.

Previous research has documented that *Orobanche* spp. and *Phelipanche* spp. have a complex genetic structure due to their allogamous mating: *O. crenata* from Ethiopia [25] or from Algeria [28] and *Orobanche* spp. from Spain [24,33,34].

Our broomrape population structure investigation showed that, at K = 2, the model-based clustering divides the studied samples into two subgroups, the first of which grouped *O. crenata* and *P. ramosa* together, and the second of which included samples from all of northern Tunisia without showing any particular correlation with the geographic origin of the samples or overlapping between the two groups. Conversely, when we move to K = 3, we can distinguish three groups, and *O. foetida* is clearly distinct from the two other species. This could be explained by the fact that an outburst of *Orobanche* races from wild to cultivated species has been documented, as reported by many researchers in the Mediterranean regions [5,29,33]. For instance, *O. foetida* was reported in Tunisia in 1905 on *Medicago truncatula* [50]; and also on common vetch [51] and lentil [52] in Morocco with variable levels of parasitism. The results of [11], utilizing Radseq to explore *O. foetida* genetic variability, are in agreement with the grouping of *O. foetida* separately from the other species in this study and the high genetic variability within the population without any geographic origin correlation. Indeed, both [11,33] pinpointed the autogamous mating of Tunisian *O. foetida* populations, affecting crop plants compared with the allogamous mating of Spanish *O. foetida* that parasitizes wild species.

Moreover, we noticed during our field tours and sampling expeditions that *O. crenata* and *P. ramosa* [11] were predominant in the same regions and fields; for example, in the Kairouan region, we found *O. crenata* to be very common on milk thistle (*Silybum marianum*) in uncultivated and zero-tillage fields. In that region, farmers grow peas and faba beans in the winter season; then, they move to tomato cultivation during the summer season with no knowledge of the parasites. As such, this practice increases the differentiation process from wild to cultivated species, as both *O. crenata* and *P. ramosa* grow in the same regions and have wild hosts that keep them growing in fields offseason, and they will thus cross-pollinate.

In our study, Figure 3 shows that *O. crenata* samples are clustered together without region differentiation, in agreement with a previous study [25]. The aforementioned results, however, contradict those of [48], which discovered a distinct differentiation between Moroccan *O. crenata* accessions based on their place of origin. Nevertheless, the same results were found by [25,28] in Ethiopia with *O. crenata* populations. The clustering without correlation with sample geographic localization suggests that there is significant mixing or outcrossing in the gene pool of *O. crenata* between populations, supported by the AMOVA's high genetic diversity inside populations compared with the diversity between populations. Moreover, we noticed that the farming practices (i.e., seed exchange) helped the spread of *O. crenata* and *P. ramosa* from the north to the center. Additionally, there is always a succession from non-cropping, zero-tillage to the cultivation of two seasons with broomrape-sensitive crops (pea or faba bean cultivars in winter and tomato in summer). This practice makes the parasite not mutate very much, as there is always a susceptible host in both cases (wild species and cultivated crops). This is obvious when we closely look

at the subgroup of *O. crenata*, where populations of parasitized *Silybum marianum* from the north and central regions of the country are grouped together with two populations of parasitized faba beans from the Kairouan region. On the other hand, the *O. crenata* populations parasitizing other wild species are grouped together with two broomrape samples parasitizing two *V. faba* cultivars from the Bizerte region, where the farmers are aware of broomrape problems, and intensive weeding, crop rotation and the use of resistance cultivars are taking place, which may promote the parasite's differentiation.

## 5. Conclusions

The current study seeks to estimate the genetic variability and structure of the most devastating broomrape species in Tunisia. Indeed, RAPD-dominant markers were able to demonstrate appropriate polymorphism and provided adequate and clear information relative to the genetic diversity of *O. foetida*, *O. crenata* and *P. ramose* and their populations' structures given the lack of full genome sequencing.

A significant genetic disparity within individuals of each genus and species resulted in us classifying the Tunisian *Orobanche* spp. and *Phelipanches* spp. into two main metapopulations and then into two genetic groups based on genius and species diversity levels, deprived of a geographic origin correlation. The low levels of diversity between the populations indicate that breeding schemes for rendering resistance to grain legumes against broomrapes can be conducted in one location. The present study is original and a baseline for studying the diversity and population structure of two genera of Tunisian broomrape, *Orobanche* spp. and *Phelipanche* spp. An additional screening based on the available markers of each species, such as ISSR, SSR and plastid DNA polymorphism, and via high throughput techniques such as HRM and GBS, with a large sample from the neighboring Mediterranean countries, would bring better knowledge and understanding about the diversity and population structure to assist breeding for resistance.

**Author Contributions:** Conceptualization, K.K., E.T., R.C. and M.K.; methodology, K.K., Z.A., E.T. and A.K.; software, A.K., K.G. and M.R.; validation, K.K., E.T. and M.K.; formal analysis, K.K., Z.A., E.T., A.K., S.K. and K.G.; investigation, K.K., Z.A., E.T., A.K., S.K. and K.G.; resources, R.C. and M.K.; author data curation, K.K., Z.A., E.T., A.K., S.K. and K.G. writing—original draft preparation, K.K., Z.A., E.T., A.K. and K.G.; writing—review and editing, K.K., Z.A., E.T., R.C., D.C. and M.K.; visualization, all authors; supervision, R.C. and M.K.; project administration, D.C. and M.K.; funding acquisition, D.C. and M.K. All authors have read and agreed to the published version of the manuscript.

**Funding:** The project was partially funded by PRIMAII ZEROPARASITIC Project, PRIMA 2018, Section 2, "Innovative sustainable solutions for broomrapes: prevention and integrated pest management approaches to overcome parasitism in Mediterranean cropping systems".

**Institutional Review Board Statement:** Not applicable.

**Informed Consent Statement:** Not applicable.

**Data Availability Statement:** Data can be available upon request.

**Acknowledgments:** The authors wish to thank the Ministry of Agriculture, Hydraulic Resources and Maritime Fisheries; the Ministry of Higher Education and Scientific Research. In addition, we would like to thank Ioanna Kendrick for English revisions of an earlier draft of the manuscript and Maria Gerakari for helping with proofreads.

**Conflicts of Interest:** The authors declare no conflict of interest.

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
