# Peer review of "Genetic Structure and Diversity Analysis of Tunisian Orobanche spp. and Phelipanche spp. Using Molecular Markers"

_applsci, doi:10.3390/app132111622_

Round 1
Reviewer 1 Report
In this paper, the authors addressed a critical agricultural issue related to parasitic plants, specifically Orobanche and Phelipanche species, which can cause substantial economic losses. The study focuses on these parasitic species in Tunisia, where they pose a significant threat to important crops. The need for early detection and effective management of these parasitic plants is highlighted, as they can cause severe damage before their above-ground appearance. To address this issue, the paper employs various molecular markers to evaluate the genetic diversity of Orobanche and Phelipanche populations in Tunisia, aiming to understand their genetic structure and variability. This is an interesting paper, however, i have a few concern that should be addressed before it can be accepted for publication.
1. Line 45-52 - The paper introduces Broomrapes but could be more explicit about the primary research problem or hypothesis. What specific question or objective is the study addressing.
2. Line 56-57 - Can you provide the source for the information on the hectares of legume crops infected by broomrape? Including additional statistics on economic losses would strengthen the argument.
3. Line 78-81 - The importance of genetic studies is highlighted, but it could be beneficial to explain more explicitly why understanding genetic diversity and structure is crucial for breeding programs and management strategies. In this regard the authors can provide and cite some molecular markers based recent researches such as:
http://dx.doi.org/10.1007/s10722-022-01493-5
http://dx.doi.org/10.1111/pbr.13016
Providing a brief explanation of the different types of markers (RAPD, ISSR, AFLP, SSR, SNPs) and their applications could be helpful.
4. Line 112-115 - Sample Selection and Geographic Localization: It's important to provide more context regarding why these specific populations of O. crenata, O. foetida, and P. ramosa were chosen for the study.
5.Line 224-225 - Primer OPD20: You mention that primer OPD20 had 88% polymorphism, which is lower than the other primers. Can you discuss why this primer had a lower polymorphism rate, and could this affect the interpretation of the results?
6. Line 257 - Genetic Closeness: You mention genetic closeness between P. ramosa and O. crenata. Can you provide some insights or hypotheses about why this closeness exists? Is it related to their geographical distribution or host preferences?
7. Line 279 - ΔK Values: You mention ΔK values (ΔK = 139.06 at K = 2 and ΔK = 6.66 at K = 4). Could you briefly explain what ΔK represents and its significance in determining the optimal number of clusters? How does this help in understanding the genetic structure of Tunisian broomrapes?
Minor corrections are required
Author Response
Thank you very much for each fruitful comments that helped us improving our manuscript.
Please find attached a step by step answer to each comment.
Kind regards
The corresponding authors.

Reviewer 2 Report
Abstract
It is mentioned that it is important to know the genetic diversity of boomrapes for an eventual breeding program )lines 27-28). Why should genetic improvement be done in parasitic species? Do you mean genetic improvement of crops rather than parasitic plants? Clarify.
Materials and Methods
Was DNA extracted indvidually or was the tissue/DNA of the individuals in each accession mixed? How many individuals were used from each accession to extract DNA and perform the analysis?
RAPDs were important as molecular markers several decades ago because there were no other satisfactory options, but these were discontinued due to their well-known reproducibility problems; also, their dominant nature gives them little power to estimate genetic diversity. Why were these markers used when virtually all the options that exist are better? Why were these markers used when virtually all the options that exist are better? Justify.
Results
Information in lines 216-217 corresponds to materials and Methods section. The first section of Results (lines 215-240), including Table 6, is devoted to describing characteristics of primers and markers themselves, which is not part of the objective. It is suggested to delete this section.
Explain the meaning of GTS, so that it is evident that the author understands and applies the concept correctly.
For the benefit of readers, explain and contrast the concepts of Na and Ne as mentioned in Table 7.
Conclusiones
The information contained in lines 378-382 does not correspond to conclusions, since the description of the markers was not part of the objective of the study.
Lines 387-389. Information not supported by study results.
The authors failed to establish a connection between the aggressiveness of the populations of Orobanche and their genetic diversity, as they had stated in their objectives.
English language writing should be reviewed by an expert or native speaker of that language.
Author Response

(The authors gave the same response as above.)

Round 2
Reviewer 2 Report
The authors made important changes to substantiate some of the study’s weaknesses; however, some aspects can still be noted:
1. The main drawback of RAPDs is their frequent lack of repeatability and their low power for genetic diversity estimates, given their dominant nature.
2, It is not necessary to rely on the full genome of orobanche (as the author argues) to use more robust markers such as SSRs.
3. In Materials and Methods, the authors should make clear the number of individuals involved in the analysis.
4. If the authors insist on concluding on aspects related to the molecular markers used, then they should include this point as one of the objectives of the study
Author Response
Thank you very much for your valuable comments. Attached you will find a step by step response to all the comments.
Kind regards
The corresponding authors
